# Selective Impact of Selenium Compounds on Two Cytokine Storm Players

**DOI:** 10.3390/jpm13101455

**Published:** 2023-09-30

**Authors:** Indu Sinha, Junjia Zhu, Raghu Sinha

**Affiliations:** 1Department of Biochemistry and Molecular Biology, Penn State Cancer Institute, Penn State College of Medicine, Hershey, PA 17033, USA; isinha@pennstatehealth.psu.edu; 2Department of Public Health Sciences, Penn State Cancer Institute, Penn State College of Medicine, Hershey, PA 17033, USA; jzhu2@pennstatehealth.psu.edu

**Keywords:** methylseleninic acid, sodium selenite, selenous acid, selenomethionine, cytokines, cytokine storm, COVID-19

## Abstract

COVID-19 patients suffer from the detrimental effects of cytokine storm and not much success has been achieved to overcome this issue. We sought to test the ability of selenium to reduce the impact of two important cytokine storm players: IL-6 and TNF-α. The effects of four selenium compounds on the secretion of these cytokines from THP-1 macrophages were evaluated in vitro following an LPS challenge. Also, the potential impact of methylseleninic acid (MSeA) on Nrf2 and IκBα was determined after a short treatment of THP-1 macrophages. MSeA was found to be the most potent selenium form among the four selenium compounds tested that reduced the levels of IL-6 and TNF-α secreted by THP-1 macrophages. In addition, an increase in Nrf2 and decrease in pIκBα in human macrophages was observed following MSeA treatment. Our data indicate that COVID-19 patients might benefit from the addition of MSeA to the standard therapy due to its ability to suppress the key players in the cytokine storm.

## 1. Introduction

Selenium is an essential trace element which plays crucial roles in the health and development of several human disorders [1]. Selenium supplementation is critical in the prevention and/or treatment of a variety of disorders, including Kashin–Beck disease [2], hypothyroidism [3,4], atherosclerosis [5], cardiovascular disease [6], Alzheimer’s disease [7], HIV and AIDS [8], and more recent indications are emerging for COVID-19 [9,10,11]. Furthermore, several selenium forms have been investigated in different cancer types for their potent anticancer activity, including but not limited to lung, breast, prostate, colon cancer, cervical, bladder and pancreatic cancer [12,13,14,15,16,17,18,19]. 

During viral infections, several reactive oxygen species (ROS) are produced, and these could be beneficial as well as harmful for a variety of cellular functions [20,21]. Viral replications can also be enhanced due to these ROS, and because of the viral infection, there is usually a demand for micronutrients in the host environment, which could cause their deficiency. Selenium plays a vital role in antioxidant defense mechanisms, influencing redox signaling and redox homeostasis. Selenium deficiency, which is the main regulator of selenoprotein expression, has been associated with the pathogenicity of several viruses. In addition, several selenoproteins, including glutathione peroxidase (GPX) and thioredoxin reductase (TXNRD), seem to be playing important roles in various models of viral replication. 

Selenium has also been implicated in systemic inflammatory response syndrome, sepsis, and septic shock [22,23,24] and could overcome adverse effects and reduce mortality rates in severe sepsis [23]. In a sepsis-related lung injury, selenium in combination with niacin caused synergistic activation of the glutathione redox cycle, a reduction in hydrogen peroxide levels and upregulation of nuclear factor erythroid 2-related factor 2 (Nrf2) and improved survival in rats [24]. 

In recent years, it has become clear that increased levels of cytokines such as IL-1, IL-2, IL-6, GMCSF, IFN-γ and TNF-α in COVID-19 patients can lead to cell death, tissue damage and fever, and can impact vascular physiology and coagulation [25]. Regarding cytokines, the earliest report on Tocilizumab, an anti-IL-6 receptor used for the treatment of patients with COVID-19-related lung disease [26], opened possibilities for severe cases. A recent meta-analysis also revealed that IL-6 is a master regulator of COVID-19 severity biomarkers [27]. In addition, severe acute respiratory syndrome coronavirus 2 (SARS-CoV-2) spike protein S1 subunit activates TLR4 signaling to induce pro-inflammatory responses in murine and human macrophages. Therefore, TLR4 signaling in macrophages may be a potential target for regulating excessive inflammation in COVID-19 patients [28].

Regional selenium deficiency might be related to an increased fatality rate of COVID-19 in China [29]. To date, most observation studies indicate that a lower serum selenium level is associated with worse outcomes [30,31,32]. In a recent study, low selenium and zinc levels measured according to intake and toenail concentrations revealed an inverse relationship with increasing COVID-19 severity index in young adults [33]. The severity index is calculated based on ten commonly reported symptoms and their duration in COVID-19 infection. More severe selenium deficiency has also been proposed to correlate with COVID-19 disease progression, and furthermore, selenium deficiency is associated with overall organ system dysfunction and death [34]. To our knowledge, no randomized clinical trials have appropriately investigated the effect of selenium supplementation on COVID-19. A clinical trial in Spain is investigating whether daily micronutrient supplementation with 110 μg of selenium along with 10 other vitamins and minerals for 14 days in 300 adults with COVID-19 reduces the need for hospitalization due to the disease [35]. Another trial is examining the effects of 2000 μg of selenium (as a selenous acid infusion) on day 1 followed by 1000 μg on days 2–14 plus standard-of-care therapy in 100 hospitalized adults with moderate, severe or critical COVID-19 [36]. However, a few studies on the effects of supplementation with the second most important trace element, zinc, in COVID-19 did not confirm its efficacy [37]. 

For the current study, we planned to evaluate the effects of different selenium compounds on key players of cytokine storm that are critical to target in COVID-19 patients. Since the lipopolysaccharide (LPS)-induced expression profile of THP-1 cells was found to be like that of human PBMC-derived macrophages [38], we induced inflammation with LPS in THP-1 macrophages to study the impact of selenium compounds on IL-6 and TNF-α. Our results are strongly encouraging for follow-up research on selective selenium compounds in all the cytokines involved in the cytokine storm for COVID-19 patients and show potential for using selenium as an adjuvant therapy in long haulers of COVID-19 infection. 

## 2. Materials and Methods

### 2.1. Selenium Compounds 

Selenous acid (SA), methylseleninic acid (MSeA), sodium selenite (Sel) and selenomethionine (SM) were purchased from Sigma (St. Louis, MO, USA) and used at concentrations ranging from 0.25 μM to 10 μM for treatments. 

### 2.2. THP-1 Cell Line, THP-1 Macrophages, LPS Challenge and Selenium Treatments 

The THP-1 cell line was obtained from The American Type Culture Collection (ATCC, Manassas, VA, USA). The details on genetic information along with specific markers for THP-1 cell line are provided in the Cellosauras database (accession CVCL_0006). These cells were cultured in RPMI 1640 (ATCC, Manassas) with 10% heat-inactivated FBS (Gemini Bio-Products, West Sacramento, CA, USA) and 1% Penn-Strep (Invitrogen, Carlsbad, CA, USA). These cells were maintained and treated at 37 °C in a humidified incubator in the presence of 5% CO_2_. The cells were maintained in culture for only 4–5 passage numbers for a given experiment. Short Tandem Repeat (STR) DNA profiling was used to confirm the authenticity of the cell type. THP-1 cells in duplicate wells were differentiated into macrophages (THP-1 macrophages) with 100 nM Phorbol 12-myristate 13-acetate (PMA; Sigma, St. Louis, MO, USA) for 24 h, and the next day, the plate was rinsed twice in PBS and replenished with regular medium and the cells were given a rest for another 24 h in regular medium. The THP-1 macrophages were then challenged with LPS (100 ng/mL) in the absence and presence of selenium compounds for 24 h. The spent media from the treated cells were collected and stored at −20 °C until used for IL-6 and TNF-α measurements, and the adherent cells were processed for an MTT assay. 

### 2.3. Cytotoxicity (MTT Assay) 

Selenium-treated THP-1 macrophages were incubated with 3-(4,5-dimethylthiazol-2-yl)-2,5-diphenyltetrazolium bromide (MTT, Sigma, St. Louis, MO) (500 μL of 5 mg/mL solution) for 3 h in dark at 37 °C. The MTT solution was removed from the wells and replaced with 500 μL of DMSO/well to dissolve the purple-blue formazan particles, and the plate was read at 570 nm with correction at 630 nm in a Molecular Devices plate reader. An MTT assay was performed in duplicate for each selenium compound. 

### 2.4. Evaluating Inflammatory Response Markers: IL-6 and TNF-α 

The frozen spent medium from selenium-treated THP-1 macrophages was thawed, and the secreted IL-6 and TNF-α levels were estimated using Quantikine^TM^ ELISA Kits (R&D Systems, Minneapolis, MN, USA) following the manufacturer’s instructions. IL-6 and TNF-α assays were performed in duplicate following treatments with each of the selenium compounds. 

### 2.5. Western Blotting 

THP-1 cells were differentiated into macrophages with PMA as described above, and MSeA (0, 5, 10 μM) was incubated in the presence or absence of LPS for 30 min. The cells were collected, washed in cold PBS and lysed in RIPA buffer (Sigma, St. Louis, MO, USA) containing protease inhibitors. Equal amounts of protein (50 μg) were subjected to SDS-PAGE, and Western blot was performed on the proteins transferred onto nitrocellulose membrane, as described earlier [39]. Primary antibodies against pIκBα, IκBα and Nrf2 (Cell Signaling, Danvers, MA) (1:1000 dilution) and β-actin (Proteintech, Chicago, IL, USA) (1:10,000 dilution) were reacted separately with the blot. The HRP-conjugated anti-rabbit and anti-mouse secondary antibodies (Cell Signaling, Danvers, MA, USA) were incubated at a dilution of 1:3000. Band expressions were developed using Pierce^TM^ ECL reagents (Thermo Scientific, Rockford, IL, USA). 

### 2.6. Statistical Analysis

Descriptive statistics were generated for the continuous outcome variables (IL-6, TNF-α levels and MTT assay) for each selenium compound at various concentrations. The main comparisons were performed between the LPS-alone treatment and each selenium compound at different concentrations. Dunnett’s tests were used on the pairwise comparisons between the LPS alone and other selenium groups. Due to the exploratory nature of this study, the statistical significance level was not further adjusted for multiple testing. All analyses were performed using the statistical programming language R version 4.3.1 (The R Foundation for Statistical Computing). All tests were two-sided, and the statistical significance level used was 0.05.

## 3. Results

### 3.1. Moderate Effect of Selenium Compounds on Cytotoxicity 

LPS alone (100 ng/mL) caused some cell death in THP-1 macrophages in our set up (data not shown). To evaluate the cytotoxicity of selenium compounds in these LPS treated THP-1 macrophages, an MTT assay was performed following 24 h treatments. MSeA at all doses was able to rescue the macrophage cell death caused by LPS (Figure 1). However, significant cytotoxicity was observed in SM (10 μM, *p* < 0.01) treated cells as compared to LPS alone while only moderate cytotoxicity (*p* > 0.05) was observed following treatments of THP-1 macrophages with Sel and SA at higher doses.

### 3.2. MSeA, Sel and SA Effectively Reduce IL-6 Levels 

The levels of LPS-induced TNF-α secreted by THP-1 macrophages were inhibited by 11%, 48% and 75% at 2.5 μM, 5 μM and 10 μM (*p* < 0.001), respectively (Figure 2). By contrast, Sel slightly (2%) but significantly inhibited TNF-α levels at 2.5 μM and 5 μM (*p* < 0.05), but SA and SM did not impact the TNF-α levels secreted at the doses tested. The levels of TNF-α secreted following selenium treatments were not influenced by any direct impact on cell viability, as shown in Figure 1.

### 3.3. MSeA Drastically Reduces TNF-α Levels 

The LPS-induced TNF-α levels secreted from THP-1 macrophages were inhibited by 11%, 48% and 75% at 2.5 μM, 5 μM and 10 μM (*p* < 0.001) respectively (Figure 3). By contrast Sel slightly (2%) but significantly inhibited TNF-α levels at 2.5 μM and 5 μM (*p* < 0.05) but SA and SM did not impact the TNF-α levels secreted at the doses tested. The levels of TNF-α secreted following selenium treatments were not influenced by any direct impact on cell viability as shown in Figure 1.

### 3.4. MSeA Influences Nrf2 and IκBα Levels 

Following a 30 min treatment of THP-1 macrophages, MSeA strongly increased Nrf2 levels in the absence of LPS (2.7- and 3.4-fold for 5 and 10 μM, respectively) and only moderately increased it in the presence of LPS (2.2-fold for both 5 and 10 μM) (Figure 4). LPS alone reduced the Nrf2 levels in THP-1 macrophages (0.6-fold). Furthermore, the pIκBα levels were diminished (0.4- and 0.02-fold for 5 and 10 μM, respectively) by MSeA in the absence of LPS and moderately decreased (0.7- and 0.6-fold for 5 and 10 μM, respectively) in the presence of LPS. On the other hand, the native IκBα levels were elevated in the absence of LPS and reduced in its presence; however, MSeA did not change these levels in the presence or the absence of LPS (Figure 4). Since LPS induces phosphorylation (5.6-fold), the native IκBα will be degraded in the presence of LPS, as is evident from the results. This in turn would allow NF-κB to go into the nucleus. Without LPS, however, native IκBα is intact and therefore not degraded. Thus, we observed higher native IκBα levels in the absence of LPS as compared to the native IκBα levels in the presence of LPS. Figure 5 describes these results in a schematic diagram. 

## 4. Discussion

Several recent reviews in the literature have cited the potential role of selenium in SARS-CoV-2 infection [9,10,11,40]. However, a recent study showed that daily intravenous selenium (1 mg Sel) in a cohort of COVID-19 patients with severe ARDS for a total of two weeks elevated their selenoprotein P and GPX3 levels [41]. Furthermore, the selenoprotein P levels were inversely proportional to the levels of inflammatory cytokines CRP, IL-6 and IL-1β. A caveat in this study was that the patients had also received different combinations of artificial nutrition which contained varied amounts of selenium and zinc, thus making it more challenging to infer whether the impact on the patients was solely due to selenium. Others have proposed evaluating a series of selenium-based compositions (patents pending) to fight against COVID-19 [42], and it would be worth investigating if any of these selenium forms influence the components of the cytokine storm. 

The SARS-CoV-2 spike protein S1 subunit activates TLR4 signaling, which in turn induces a pro-inflammatory response in murine and human macrophages. Thus, TLR4 signaling in macrophages could be a potential target for regulating excessive inflammation in COVID-19 patients [28]. In our current study, we used LPS as the TLR4 agonist in THP-1 cells after differentiation via PMA treatment. Using these relevant THP-1 macrophages, the potential impact of selenium on the release of IL-6 and TNF-α was measured following stimulation with LPS in the presence of several selenium compounds. 

We chose four prominently researched selenium compounds, MSeA, Sel, SA and SM. MSeA has been proposed to accumulate in SARS-CoV-2-infected cells and inactivate the M^pro^ of SARS-CoV-2 in these cells by modifying the Cys145 residue of the protease [9]. Sel is an acceptable inorganic selenium form showing promising effects on several viral infections in human studies [43]. SA was proposed as a potent selenium compound for the treatment of moderately ill, severely ill and critically ill COVID-19 patients [36]. SM was recently evaluated in an herbal combination against human beta coronavirus in vitro and in vivo [44]. In our study, MSeA, Sel and SA showed a reduction in IL-6 levels, while MSeA and to a minor extent Sel decreased TNF-α levels in the THP-1 macrophages in the presence of LPS. It was clear in these experiments that the impact of selenium compounds varied based on the form of selenium selected. However, the impact of selenium was independent of any cytotoxicity in THP-1 macrophages.

Previously, we had observed changes in the cytokine profile of healthy men supplemented with selenized yeast. The inflammatory cytokines, such as IL-6, IL-8, MCP-1, IL-12p70, IFN-γ and TGF-β, were reduced in the selenized-yeast group as compared to the placebo group [45]. However, the major form of selenium in selenized-yeast is SM, which only had a minor impact on IL-6 in our current in vitro study. An earlier report showed that Sel (2–8 µM) limited the gene expression of the pro-inflammatory cytokines IL-1β, TNF-α and IL-6 within 6 h under inflammatory conditions induced by *Staphylococcus aureus*. Further, these observations, along with the suppression of phosphorylation of IκBα and p65 at 4 μM and 8 μM Sel substitution [46] would in part indicate suppression of the inflammatory response by the NF-κB pathway. Similarly, in the present study, we observed a 40% decline in the phosphorylation of IκBα levels in MSeA-treated THP-1 macrophages following LPS induction. A recent review and meta-analysis [47] suggested that oral and intravenous selenium supplementation influences IL-6 and TNF-α levels.

On the contrary, the effects of selenium on NF-kB are controversial depending on the dose and cell type (murine vs. human). The most cited report of selenium’s effect on NF-κB was performed in RAW macrophages by first creating a selenium deficiency, which resulted in the activation of NF-κB in the presence of LPS and was reduced with the addition 250 nM Sel [48]. Meanwhile, results from a recent study in THP-1 macrophages showed that 50 nM Sel facilitates LPS-induced NF-κB activation [49].

Our preliminary study also revealed that MSeA increases Nrf2 levels and decreases pIκBα levels in the absence of LPS in THP-1 macrophages. Previously, an organic form of selenium has been shown to induce Nrf2 levels in the lungs [50]. SARS-CoV-2 infection in both in vitro and in vivo models showed a downregulation of Nrf2 protein levels and Nrf2-dependent gene expression in human airway epithelial cells as well as in the lungs of BALB/c mice. This finding clearly suggests that the therapeutic strategies for COVID-19 may consider the use of pharmacologic agents that are known to boost the expression levels of cellular NRF2. More recently, it has been revealed that SARS-CoV-2 infection strikingly decreases the activation of the Nrf2/HMOX1 axis [51], suggesting NRF2 is an important target. In the context of COVID-19, cytokine storm in severe cases decreases endothelial cell antioxidant defense via downregulation of Nrf2, which may be IL-6-dependent [52]. In addition, several Nrf2-interacting nutrients, such as berberine, curcumin, epigallocatechin gallate, genistein, quercetin, resveratrol and sulforaphane, which all function similarly to endothelial damage, lung injury and cytokine storm, might be helpful to COVID-19 patients [53]. We strongly believe that MSeA may be beneficial against the cytokine storm via a similar mechanism.

Alternatively, selenium levels in populations could be improved by supplementing soil with selenium-enriched fertilizers that kept mortality rates due to COVID-19 low in Finland [54]. The nationwide supplementation of fertilizers with sodium selenate has been shown to be an effective and safe method of increasing the selenium intake of the entire population. Improving the selenium status via nutritional measures or supplementation may be helpful in reducing the devastation caused by COVID-19, especially in areas affected by selenium deficiency worldwide [55]. Moreover, it is important to consider selenium as a cofactor to achieve a more effective immune response to COVID-19 vaccination [56].

In summary, our results indicate that MSeA reduces IL-6 and TNF-α levels in THP-1 macrophages. COVID-19 patients might benefit from supplementation with selenium forms such as MSeA due to their ability to suppress key players in the cytokine storm when used alongside the standard therapy. It is also worth exploring the use of selenium in building a better defense in COVID-19 long haulers. Future studies will focus on investigating the impact of MSeA on several players and their cell surface receptors involved in the cytokine storm caused by COVID-19 infections, including but not limited to IFN-γ, IFN-α2, IL-2, 4, 7, 10, 12 and 17, as well as chemokines such as IP-10, the macrophage-colony-stimulating factor (M-CSF) and G-CSF in THP-1 macrophages.

## Figures and Tables

**Figure 1 jpm-13-01455-f001:**
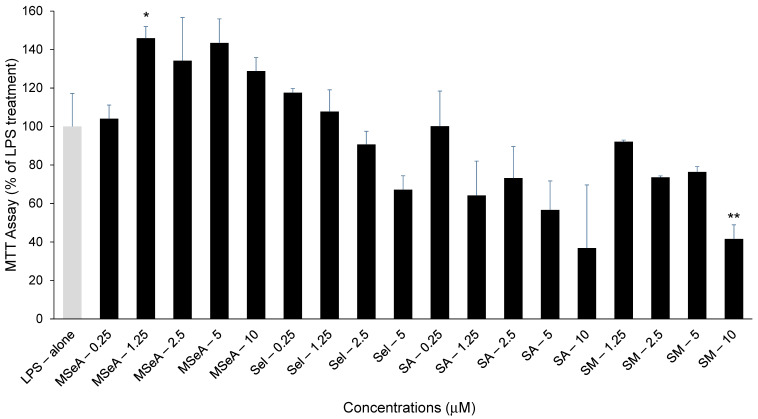
MTT Assay for selenium-treated THP-1 macrophages in presence of LPS. MSeA rescued the cells from LPS induced cell death while SM showed cytotoxicity at 10 μM. (N = 2) LPS: lipopolysaccharide, MSeA: methylseleninic acid, SA: selenous acid, Sel: sodium selenite, SM: selenomethionine. Comparisons were made between LPS alone and each selenium compound at all concentrations. * *p* < 0.05, ** *p* < 0.01.

**Figure 2 jpm-13-01455-f002:**
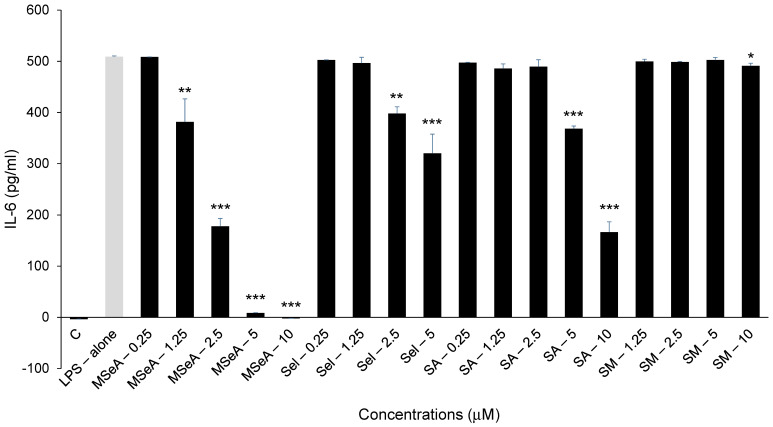
LPS-induced IL-6 levels in selenium-treated THP-1 macrophages. The IL-6 secretions from selenium-treated cells were decreased in a dose-dependent manner in most of the compounds studied. However, MSeA was the most potent among all (N = 2). C: control, LPS: lipopolysaccharide, MSeA: methylseleninic acid, SA: selenous acid, Sel: sodium selenite, SM: selenomethionine. Comparisons were made between LPS alone and each selenium compound at all concentrations. * *p* < 0.05, ** *p* < 0.01, *** *p* < 0.001.

**Figure 3 jpm-13-01455-f003:**
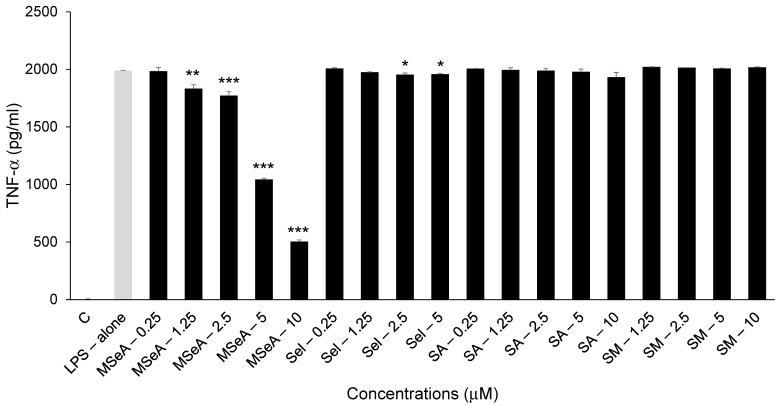
TNF-α levels in selenium-treated THP-1 macrophages. Only MSeA treatments showed a dose-dependent decrease in TNF-α levels, while the other compounds had no appreciable impact (N = 2). C: control, LPS: lipopolysaccharide, MSeA: methylseleninic acid, SA: selenous acid, Sel: sodium selenite, SM: selenomethionine. Comparisons were made between LPS alone and each selenium compound at all concentrations. * *p* < 0.05, ** *p* < 0.01, *** *p* < 0.001.

**Figure 4 jpm-13-01455-f004:**
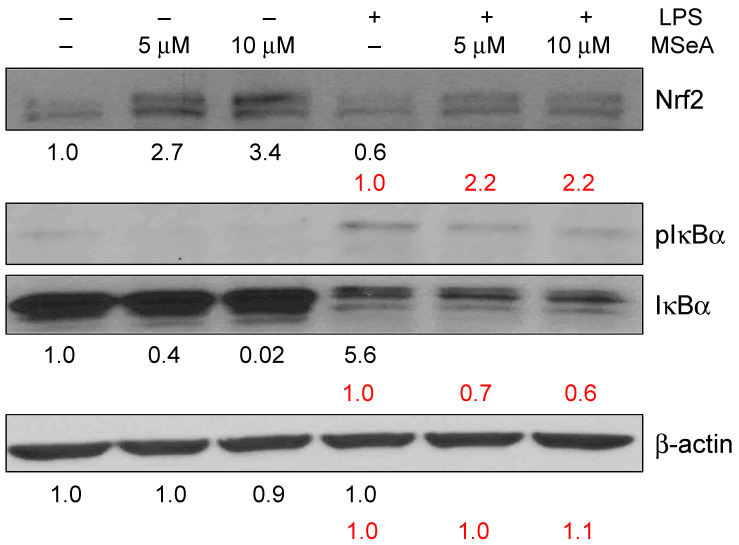
Nrf2, pIκBα and pIκBα levels in MSeA-treated THP-1 macrophages in the absence and presence of LPS. Fold changes from −LPS control are presented in black and those from +LPS control are presented in red. The band intensities were normalized by β-actin. (N = 1) LPS: lipopolysaccharide, MSeA: methylseleninic acid.

**Figure 5 jpm-13-01455-f005:**
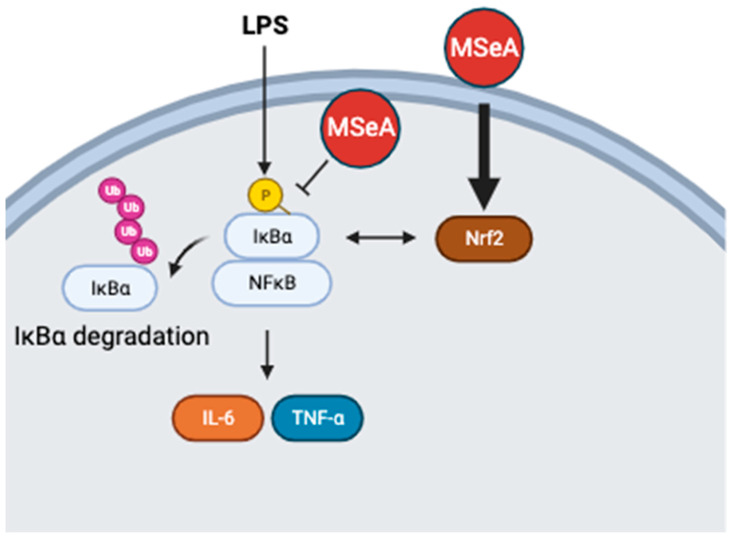
Schematic presentation of effects of LPS and MSeA on pIκBα, Nrf2 and cytokines (IL-6/TNF-α) in THP-1 macrophages. In the absence of LPS, MSeA upregulates Nrf2. In the presence of LPS, MSeA decreases pIκBα/IκBα, potentially resulting in a reduction in IL-6 and TNF-α. Schematic was created with BioRender.com.

## Data Availability

Research data are clearly depicted in the graphs and are also available to the readers upon request.

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
