# Peer review of "Selective Impact of Selenium Compounds on Two Cytokine Storm Players"

_jpm, 2023, doi:10.3390/jpm13101455_

Round 1

Reviewer 1 Report

Paper by Simha et al. presents some interesting data on the effects  on IL6 and TNF regulation by selenium treatment trough modulation of  one of NFkB activation pathway. Results are well presented but their scientific resonance appear low considering that in vivo and in vitro experiments previously published have explored these aspect. However, the in vitro experimental model used by the authors might allow to explored a large number of tips that will improve the scientific impact of the paper. I strongly suggest to analyze IL10 production, the expression on the cell surface of receptors of the three cytokines as well apoptosis and/or necrosis membrane signals. 

These aspects are less known and will increase paper importance and resonance 

Author Response

Paper by Sinha et al. presents some interesting data on the effects  on IL6 and TNF regulation by selenium treatment trough modulation of  one of NFkB activation pathway. Results are well presented but their scientific resonance appear low considering that in vivo and in vitro experiments previously published have explored these aspect. However, the in vitro experimental model used by the authors might allow to explored a large number of tips that will improve the scientific impact of the paper. I strongly suggest to analyze IL10 production, the expression on the cell surface of receptors of the three cytokines as well apoptosis and/or necrosis membrane signals. These aspects are less known and will increase paper importance and resonance 

Response: We thank the reviewer for the valuable feedback. For our pilot study (presented) we selected two of the most important inflammatory cytokines and IL-10 being anti-inflammatory was not the focus of the experiments conducted in the in vitro model. However, we will focus on other members of the cytokine storm including IL-10 in the future studies and intend to carefully investigate the cell surface receptors as well as apoptosis. We have indicated such undertaking in our discussion section. Unfortunately, conducting such experiments in a short time frame is not feasible at this point in time.

Reviewer 2 Report

This is an interesting and practical study that examines the effects of four selenium compounds on two cytokines secreted by LPS-stimulated THP-1 macrophages. However, this manuscript still contains some errors. I suggest that it should be slightly revised before it is accepted for publication.

Comment 1:Are THP-1 macrophages impacted by the presence of these four selenium compounds?

Comment 2:Please conduct a quantitative analysis of the fluorescence results.

Comment 3:To improve your article's quality and rigor, it is important to increase and enrich the experimental data. This will not only lead to a more thorough analysis but also enhance the validity and reliability of your research findings.

Comment 4:Please check the formatting and grammar of the manuscript and get a native English speaker to correct your essay.

Comment 5:Please check the references. The review must reflect the latest research on others, and if the quoted documents are obsolete references a few years ago, they can not reflect the latest research trends.

Good general English writing skills and ability to use English grammar well.

Author Response

This is an interesting and practical study that examines the effects of four selenium compounds on two cytokines secreted by LPS-stimulated THP-1 macrophages. However, this manuscript still contains some errors. I suggest that it should be slightly revised before it is accepted for publication.

Comment 1:Are THP-1 macrophages impacted by the presence of these four selenium compounds?

Response: We thank the reviewer for this question. The selenium compounds tested in THP-1 macrophages were able to impact the cytokine production based on their individual chemical nature. However, the cell viability was not significantly changed with the various treatments except for SM-5uM. In fact MSeA enhanced the cell proliferation and yet reduced the cytokine release. We have included information on the impact on cell viability on both IL-6 and TNF-a levels in the four selenium compounds investigated.

Comment 2:Please conduct a quantitative analysis of the fluorescence results.

Response: We thank the reviewer for this important comment. We have quantified the western blots and the data are expressed as fold change compared to -LPS controls and +LPS controls, this information has been included in Figure 4.

Comment 3:To improve your article's quality and rigor, it is important to increase and enrich the experimental data. This will not only lead to a more thorough analysis but also enhance the validity and reliability of your research findings.

Response: We thank the reviewer for the valuable suggestion. This was a pilot study to capture the concept of selenium being able to influence important players in cytokine storm using an in vitro system and we have successfully shown that. Future studies will consider the recommendations more efficiently.

Comment 4:Please check the formatting and grammar of the manuscript and get a native English speaker to correct your essay.

Response: We thank the reviewer for the suggestion and have made the necessary modifications in the revised manuscript.

Comment 5:Please check the references. The review must reflect the latest research on others, and if the quoted documents are obsolete references a few years ago, they can not reflect the latest research trends.

Response: We thank the reviewer for the suggestion. The only reference that we can think of being commented on by the reviewer is listed in the paragraph below. The older reference is to bring out the point in dose and cell type being discussed in the context of selenium impacted NF-kB and we feel this is relevant for discussion.

On the contrary, the effects of selenium on NF-kB are controversial depending on the dose and cell type (murine vs human). The most cited report of selenium on NF-kB was performed in RAW macrophages by first creating a selenium deficiency which resulted in the activation of NF-kB in presence of LPS and was reduced by addition 250 nM Sel [48]. While results from a recent study in THP-1 macrophages showed that 50 nM Sel facilitates LPS-induced NF-κB activation [49].

Reviewer 3 Report

There are some minor comments on the manuscript. 

1. When the cells are treated with selenium compounds, then the viability of cells will also be affected. This should be taken into account while analysing figure 1 and 2 because that would influence the levels of cytokines. It should be atleast mentioned in the text. 

2. None of the figure legend describe the number of replicates used in the experiment. Atleast 3 biological replicates should be used to come to any conclusion. 

3. In figure 3, there is no description about Sel and SA in the text. It should be mentioned somewhere. 

4. In line 170 - it should be moderate. However, this line should be rewritten in a proper english. 

5. The western blot in figure 4 should be quantified as some of the bands are not clearly visible. Its crucial to quantify from 3 independent biological replicates. 

6. In figure 4, the treatment of LPS induces phosphorylation of IkBa, but the total IkBa should be the same. Or it should be explain better in the text. It is not clear what the authors are describing from the native IkBa levels. 

7. The box is not necessary in figure 4 and the unit for the concentration of MSeA should be checked again the figure. 

8. A schematic diagram should be provided with figure 4 for a better understanding for readers. 

Some grammatical errors in the text should be corrected. 

Author Response

There are some minor comments on the manuscript. 

  1. When the cells are treated with selenium compounds, then the viability of cells will also be affected. This should be taken into account while analysing figure 1 and 2 because that would influence the levels of cytokines. It should be atleast mentioned in the text. 

Response: We are thankful to the reviewer for the comment. We have addressed the results described in figures 1 and 2 keeping in consideration any changes in the viability of cells following treatments.

  1. None of the figure legend describe the number of replicates used in the experiment. Atleast 3 biological replicates should be used to come to any conclusion. 

Response: We thank the reviewer for the comment. We have now included the number of number of biological replicates in the figure legends.

  1. In figure 3, there is no description about Sel and SA in the text. It should be mentioned somewhere. 

Response: We thank the reviewer for the comment. We have included the results for Sel and SA.

  1. In line 170 - it should be moderate. However, this line should be rewritten in a proper english. 

Response: We thank the reviewer for catching the error. We have re-written the sentence as suggested.

  1. The western blot in figure 4 should be quantified as some of the bands are not clearly visible. Its crucial to quantify from 3 independent biological replicates. 

Response: We thank the reviewer for the suggestion. We have indicated the fold-change in respective markers following MSeA treatments. The experiment was performed once (N=1) and is indicated in the figure legend). At this point in time we apologize that it is not possible to repeat the experiments as some of the reagents are backordered for several months.

  1. In figure 4, the treatment of LPS induces phosphorylation of IkBa, but the total IkBa should be the same. Or it should be explain better in the text. It is not clear what the authors are describing from the native IkBa levels. 

Response: We thank the reviewer for this question. We have clarified the point on why the native IkBa levels will not change in -LPS treated cells as opposed to +LPS cells in the results section.

  1. The box is not necessary in figure 4 and the unit for the concentration of MSeA should be checked again the figure. 

Response: We thank the reviewer for the suggestion. We have removed the box from figure 4 and have confirmed the concentrations of MSeA to be 5 and 10 uM for both -LPS and +LPS conditions.

  1. A schematic diagram should be provided with figure 4 for a better understanding for readers. 

Response: We thank the reviewer for the suggestion. We have now included Figure 5 showing a schematic for the results presented in figure 4.

Round 2

Reviewer 2 Report

The author has completed the changes as requested and I have no further comments